# Human-like AI and the Sentient Web

by Dave Raggett, (W3C/ERCIM)

**Human-like general purpose AI will dramatically change how we work, how we communicate, and how we see and understand ourselves. It is key to the prosperity of post-industrial societies as human populations shrink to a sustainable level. It will further enable us to safely exploit the resources of the solar system given the extremely harsh environment of outer space.**

Human-like AI seeks to mimic human memory, reasoning, feelings and learning, inspired by decades of advances across the cognitive sciences, and over 500 million years of neural evolution since the emergence of multicellular life. Human-like AI can be realised using functional models suitable for conventional computer hardware, complementing Deep Learning with artificial neural networks. Computational approaches to cognitive science hold huge potential for practical applications, as well as to the generation of fresh insights for ongoing research studies.

The goal of research on human-like AI is to enable cognitive agents that are knowledgeable, general purpose, creative, collaborative, empathic, sociable and trustworthy. Metacognition and past experience will be used for reasoning about new situations. Continuous learning will be driven by curiosity about the unexpected. Humans are like prediction machines, constantly seeking to understand regularities in the world around us, and directing attention to things that stand out as unexpected. Cognitive agents will be self-aware in respect to current state, goals and actions, as well as being aware of others in respect to their beliefs, desires and intents (i.e. embodying a theory of mind). Cognitive agents will be multilingual, interacting with people using their own languages.

Human-like AI will catalyse changes in how we live and work, supporting human-machine collaboration to boost productivity, either as disembodied agents or by powering robots to help us in the physical world and beyond. Of particular note are personal agents that help people with cognitive or physical disabilities – which in practice means most of us when we are old.

These changes will involve re-engineering capitalism in the post-industrial era. To see why, imagine a world where AI is widely used to replace humans to boost business profits. Taken to the extreme, the economy would collapse as there would be very few people well off enough to consume the products and services produced by businesses. For a different perspective, AI holds the key to the prosperity of post-industrial societies as human populations shrink to sustainable levels. A recent report in The Lancet predicts the population to begin declining by 2100 in nearly every country around the world. The productivity boost from AI is needed to sustain and grow the economy as the number of human workers declines. Universal income is unlikely to be the full answer, and instead we will need to explore how to redistribute the financial benefits of AI to support people in areas of interest to society as a whole.

We sometimes hear claims about existential risks from strong AI as it learns to improve itself resulting in a superintelligence that rapidly evolves and no-longer cares about human welfare. To avoid this fate, we need to focus on responsible AI that learns and applies human values. Learning from human behaviour avoids the peril of unforeseen effects from prescribed rules of behaviour, such as Azimov's three laws of robotics, and something he explored in his novels and short stories.

Human-like AI also offers the potential for extending the digital presence of each of us into a trusted personal agent (your digital self) that safeguards your privacy and personal data. For this we need to extend human rights to digital rights, and free ourselves from digital slavery for others and today's system of surveillance capitalism, see [L1]. This extension of human rights has consequences, e.g. you become liable for actions taken on your behalf by your digital self. As agents acting as digital selves improve, they will be able to learn to embody your personality, values, memories and skills, prompting the question of ownership for your digital self, which in principle, could live on after your biological death.

Human-like AI will also feature in the **Web 'verse** as distributed AR/VR that provides places to meet, to play, to learn, to do business and so much more. The Web 'verse will be populated with avatars for both humans and cognitive agents. For more details, see the 1994 paper on VR Web [L2]. This builds upon the **Sentient Web** as a federation of cognitive agents distributed across the Web with perception, reasoning and action. The Sentient Web will subsume the IoT, Web of Things and the Semantic Web. We can look forward to an evolution of Web search to become smarter and more personal.

This features pull-based e-commerce with trusted personal agents (your digital self) that work with other specialised agents on your behalf. Personal agents collate rich personal information and share directly relevant parts, subject to terms and conditions, and based upon a model of your values and preferences, as learned from your behaviour and those of others like you. Your agent conducts an auction with third parties to provide compelling offers based upon your stated needs. This gives consumers ownership over their privacy, in contrast to today's auctions for advertising space in web pages, which are based upon information obtained by surveilling your behaviour across the Web.

Human-like AI is being incubated in the **W3C Cognitive AI Community Group** [L3], along with a growing suite of web-based demos including: counting, decision trees, industrial robots, smart homes, natural language, self-driving cars, a browser sandbox and test suite, and an open source JavaScript chunks library. The approach is based upon the following cognitive architecture:

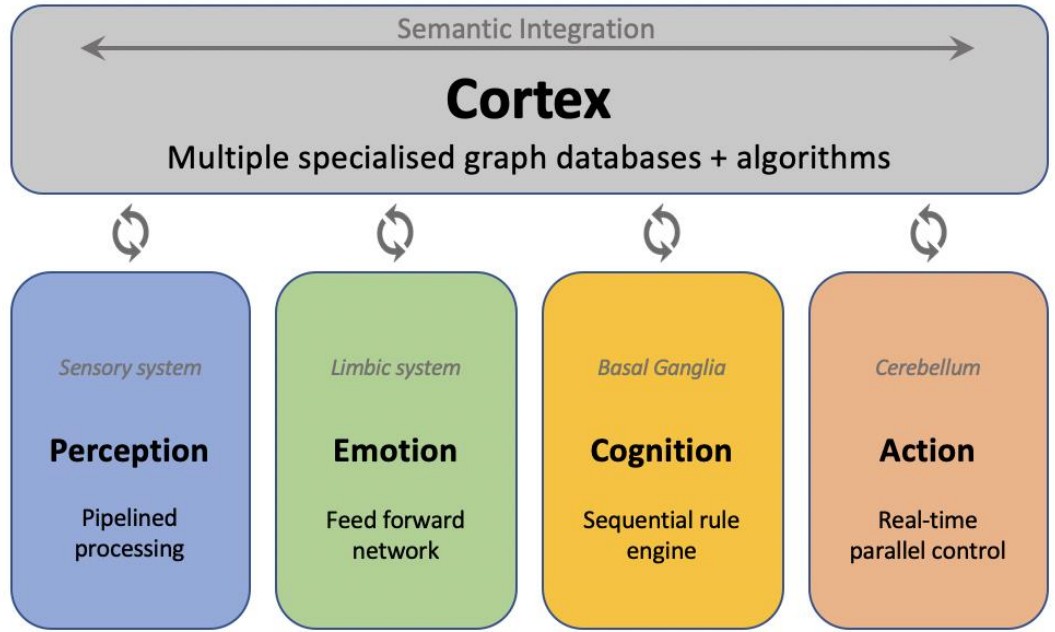

*Fig 1: Cognitive Architecture with multiple cognitive circuits equivalent to a shared blackboard*

- **Perception** interprets sensory data and places the resulting models into the cortex. Cognitive rules can set the context for perception, and direct attention as needed. Events are signalled by queuing chunks to cognitive buffers to trigger rules describing the appropriate behaviour. A prioritised first-in first-out queue is used to avoid missing events closely spaced in time.
- **Emotion** is about cognitive control and prioritising what's important. The limbic system provides rapid assessment of situations without the delays incurred in deliberative thought. This is sometimes referred to as System 1 vs System 2.
- **Cognition** is slower and more deliberate thought, involving sequential execution of rules to carry out particular tasks, including the means to invoke graph algorithms in the cortex, and to invoke operations involving other cognitive systems. Thought can be expressed at many different levels of abstraction.
- **Action** is about carrying out actions initiated under conscious control, leaving the mind free to work on other things. An example is playing a musical instrument where muscle memory is needed to control your finger placements as thinking explicitly about each finger would be far too slow.

Zooming in on cognition, we have the following architecture, which derives from work by John Anderson on ACT-R [L4, R1]. The buffers each hold a single chunk, where each chunk is equivalent to the concurrent firing pattern of the bundle of nerve fibres connecting to a given cortical region. This works analogous to HTTP, with buffers acting as HTTP clients and the cortical modules as HTTP servers. The rule engine sequentially selects rules matching the buffers and either updates them directly or invokes cortical operations that asynchronously update the buffers.

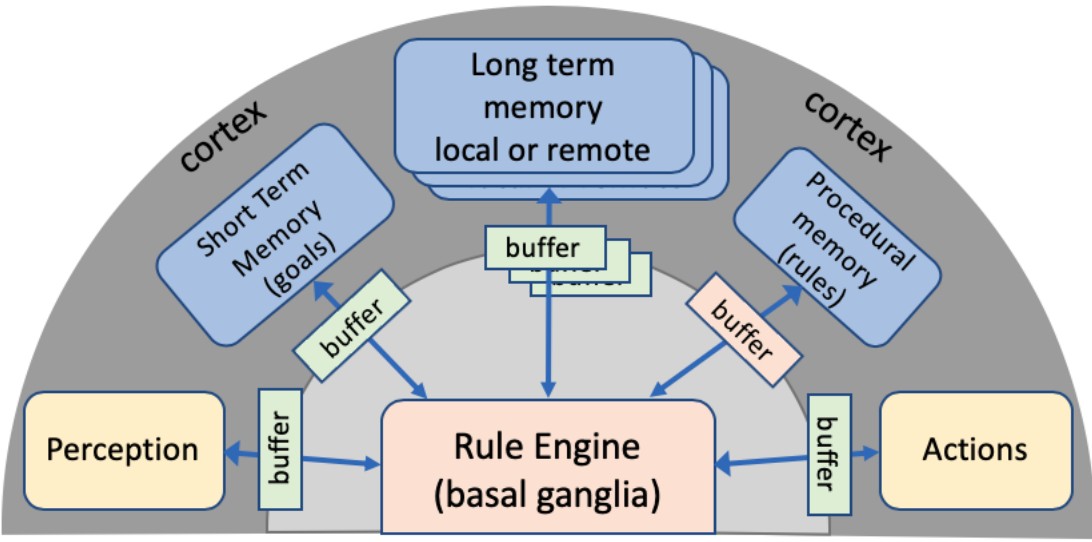

*Fig 2: Cortico-basal ganglia circuit for cognition*

A lightweight syntax has been devised for chunks, treating them as a collection of properties whose values are literals or names of other chunks, and equivalent to $n$-ary relationships in RDF. Chunks provide a combination of symbolic and sub-symbolic approaches, with graphs + statistics + rules + algorithms. Chunk modules support stochastic recall analogous to Web search. Chunks enable explainable AI/ML and learning with smaller datasets using prior knowledge and past experience.

Symbolic AI suffers from the bottleneck caused by reliance on manual knowledge engineering. To overcome this challenge, Human-like AI will mimic how children learn in the class room and in the playground. Natural language is key to human-agent collaboration, including teaching agents new skills. Human languages are complex yet easily learned by children [R2], and we need to emulate that for scalable AI. Semantics is represented as chunk-based knowledge graphs in contrast to Computational Linguistics and Deep Learning, which use large statistics as a weak surrogate. Human-like AI doesn't reason with logic or statistical models, but rather with mental models of examples and the use of metaphors and analogies, taking inspiration from human studies by Philip Johnson-Laird [L5, R3].

Humans are good at mimicking each other's behaviour with increasing practice yielding increasing skill. An example is babies learning to smile socially at the age of 6-12 weeks. Language involves a similar process of mimicry with shared rules and statistics for generation and understanding. Work on cognitive natural language processing is focusing on the end-to-end communication of meaning, with constrained working memory and incremental processing, avoiding backtracking through concurrent processing of syntax and semantics.

Work on human-like AI is still in its infancy, but is already providing fresh insights as how to build practical AI systems. This is largely due to the focus on combining ideas from different disciplines rather than being stuck within a single scientific silo. It is time to give computers a human touch!

p.s. A recent talk describing the application of human-like AI to digital transformation, along with technical details and a suite of web-based demos, can be found at [L6].

**Links:**
[L1]: https://ec.europa.eu/futurium/en/content/nature-self-digital-age
[L2]: https://www.w3.org/People/Raggett/vrml/vrml.html
[L3]: https://www.w3.org/community/cogai/
[L4]: http://act-r.psy.cmu.edu/about/
[L5]: https://www.pnas.org/content/108/50/19862
[L6]: https://www.w3.org/2021/digital-transformation-2021-03-17.pdf

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

**Contact:**

Dave Raggett,

W3C/ERCIM,

dsr@w3.org
