# OpenReview forum: "Human-like AI and the Sentient Web"
_eswc-conferences.org/ESWC/2021/Conference/Poster_and_Demo_Track — Submitted to ESWC2021 P&D_

### Official Review · AnonReviewer2 · 2021-04-08
**The paper does not satisfy the submission requirements**

**Rating:** 3
**Confidence:** 5

**Review:**

The paper does not satisfy the submission requirements. However, I provide a short review.

The paper introduces and describes human-like AI. However, the paper has a divulgation style, such as a news article. It is not clear the scientific contribution. I suppose that the goal of the paper is to introduce the current work of the W3C Cognitive AI Community Group on the human-like AI topic.

**Anonymity:**

Yes, I would like my review to remain anonymous.

---

### Official Review · AnonReviewer1 · 2021-04-12
**position paper not clear enough**

**Rating:** 4
**Confidence:** 3

**Review:**

While a position paper can generally be broad, this one makes several statements that are so sweeping as to require a lot of acquiescence from an audience.

The economic consequences should be either omitted or further explored. Using AI "to replace humans to boost business profits" is not obviously compatible with exploring "how to redistribute the financial benefits of AI". You have also mentioned personal benefits (e.g. "personal agents") as an alternative world to imagine.

Key parts of the argument are not clear to me. In particular:
- The connection between perception, emotion, cognition, and action to "human-like AI" with agreement with/respect for human values is not clear.
- How Figure 2 is "analogous to HTTP" is not clear to me.

Minor issues:
page 1: Azimov is usually spelled Asimov

References
Reference books as books (e.g. no quotations, italics)

**Anonymity:**

Yes, I would like my review to remain anonymous.

---

### Official Review · AnonReviewer4 · 2021-04-15
**Intelligent web agent vision paper**

**Rating:** 4
**Confidence:** 5

**Review:**

This paper postulates as to how human-like general purpose AI will in the future change the way we work and how we communicate.

While I'm a big supporter of research that advances the intelligent web agent vision and appreciate the proposed cognitive architecture, I would be interested in a tighter coupling between the architecture and the possible supporting technology. Additionally, the paper could be strengthened if claims made in the article where backed up either by reference or strong argumentation.

Aside from the fact that the paper does not adhere to the required LNCS format, and thus does not meet the submission criteria, the paper is more suitable for a vision track.

**Anonymity:**

Yes, I would like my review to remain anonymous.

---

### Decision · Program_Chairs · 2021-04-19

Reject